# The Risks and Benefits of Immune Checkpoint Blockade in Anti-AChR Antibody-Seropositive Non-Small Cell Lung Cancer Patients

**DOI:** 10.3390/cancers11020140

**Published:** 2019-01-24

**Authors:** Koichi Saruwatari, Ryo Sato, Shunya Nakane, Shinya Sakata, Koutaro Takamatsu, Takayuki Jodai, Remi Mito, Yuko Horio, Sho Saeki, Yusuke Tomita, Takuro Sakagami

**Affiliations:** 1Department of Respiratory Medicine, Graduate School of Medical Sciences, Kumamoto University, Honjo 1-1-1, Chuo-ku, Kumamoto-shi, Kumamoto 860–8556, Japan; rmkqq751@ybb.ne.jp (K.S.); ryosato.1981@gmail.com (R.S.); sakata-1027@hotmail.co.jp (S.Sak.); jojojojojody@gmail.com (T.J.); candypinkcolor@yahoo.co.jp (R.M.); yu1980327@yahoo.co.jp (Y.H.); saeshow@wg7.so-net.ne.jp (S.Sae.); stakuro@kumamoto-u.ac.jp (T.S.); 2Department of Neurology, Graduate School of Medical Sciences, Kumamoto University, Honjo 1-1-1, Chuo-ku, Kumamoto-shi, Kumamoto 860–8556, Japan; nakaneshunya@gmail.com (S.N.); takamakt@gmail.com (K.T.); 3Department of Molecular Neurology and Therapeutics, Kumamoto University Hospital, Honjo 1-1-1, Chuo-ku, Kumamoto-shi, Kumamoto 860–8556, Japan

**Keywords:** anti-PD-1 monoclonal antibodies, anti-acetylcholine receptor (AChR) antibody, B cell, immune checkpoint blockade, immune-related adverse events (irAEs), myasthenia gravis (MG), non-small-cell lung cancer (NSCLC), nivolumab, programmed cell death ligand 1 (PD-L1), T cell

## Abstract

Background: Anti-programmed cell death 1 (PD-1) monoclonal antibodies (Abs) unleash an immune response to cancer. However, a disruption of the immune checkpoint function by blocking PD-1/PD-ligand 1(PD-L1) signaling may trigger myasthenia gravis (MG) as a life-threatening immune-related adverse event. MG is a neuromuscular disease and is closely associated with being positive for anti-acetylcholine receptor (anti-AChR) Abs, which are high specific and diagnostic Abs for MG. Methods: A 72-year-old man was diagnosed with chemotherapy-refractory lung squamous cell carcinoma and nivolumab was selected as the third-line regimen. We describe the first report of an anti-AChR Ab-seropositive lung cancer patient achieving a durable complete response (CR) to an anti-PD-1 antibody therapy. To further explore this case, we performed multiplex immunofluorescence analysis on a pretreatment tumor. Results: The patient achieved a durable CR without developing MG. However, the levels of anti-AChR Abs were elevated during two years of anti-PD-1 antibody therapy. The tumor of the subclinical MG patient had high PD-L1 expression and an infiltrated–inflamed tumor immune microenvironment. Conclusions: This study suggests that immune checkpoint inhibitors can be safely used and provide the benefits for advanced cancer patients with immunologically ‘hot’ tumor even if anti-AChR Abs are positive. Although careful monitoring clinical manifestation in consultation with neurologist is needed, immune checkpoint inhibitors should be considered as a treatment option for asymptomatic anti-AChR Ab-seropositive cancer patients.

## 1. Introduction

Monoclonal antibodies (Abs) acting against programmed cell death 1 (PD-1) such as nivolumab and pembrolizumab are a class of drugs called immune checkpoint inhibitors that inhibit the interaction between PD-1 and programmed cell death ligand 1 (PD-L1) and unleash an immune response to cancer in contrast with chemotherapies that exert direct cytotoxic effects on tumor cells. The development of immune checkpoint blockade therapy has recently led to a paradigm shift in non-small-cell lung cancer (NSCLC) treatment and dramatically changed the treatment landscape of NSCLC patients [1,2,3].

For patients with advanced NSCLC, the immune checkpoint inhibitors have shown significant and long-lasting clinical responses in addition to a more favorable toxicity profile and improved tolerability than chemotherapy, and is currently a standard of care [2,3,4,5,6,7]. However, a disruption of the immune checkpoint function caused by blocking PD-1/PD-L1 signaling can lead to imbalances in immune homeostasis and self-tolerance, which results in an unfavorable immune response to normal tissues, which are termed immune-related adverse events (irAEs) [8,9]. The irAEs that emerge with immune checkpoint blockade therapy share clinical features with autoimmune diseases. The irAEs are usually reversible. However, in rare cases, they can be severe and life-threatening [8,10,11]. In addition, as clinical experience with immune checkpoint inhibitors increases, unexpected severe irAEs have emerged in the real-world clinical practice [10,11,12,13]. Thus, elucidating mechanisms of irAEs is urgently needed to improve their early diagnosis and develop more precise treatments for irAEs [8,9].

Myasthenia gravis (MG) is an autoimmune neuromuscular disease that is characterized by muscle weakness and fatigue, and is closely associated with a positive result for the anti-acetylcholine receptor (AChR) antibody directed against the AChR at the neuromuscular junction [14]. Anti-PD-1/PD-L1 monoclonal Abs have been known to trigger the onset of MG as one of the life-threatening irAEs [8,9,15]. Anti-AChR Abs is high specific and diagnostic antibody for MG, and the positivity of anti-AChR Abs has been reported to align with the onset of MG as an irAE in cancer patients, which discourages clinicians from using immune checkpoint inhibitors for cancer patients with pre-existing anti-AChR Abs [8,15,16]. Although several studies highlight the severity of MG as an irAE and the risks of the use of immune checkpoint inhibitors for the cancer patients with pre-existing MG or subclinical MG (asymptomatic anti-AChR Ab-seropositive cancer patients), the benefits and safety of immune checkpoint inhibitors in asymptomatic patients with pre-existing anti-AChR Abs, have not been studied [9,14,15,16,17].

In this case, we show a case of anti-AChR Ab-seropositive NSCLC patients achieving a durable complete response (CR) to an anti-PD-1 monoclonal antibody therapy (nivolumab) without developing MG. To further explore this case, we performed multiplex immunofluorescence analysis on a pretreatment tumor sample. This study provides new insights into the use of immune checkpoint monoclonal Abs for cancer patients with pre-existing anti-AChR Abs.

## 2. Results

### 2.1. An Anti-AChR Antibody-Seropositive NSCLC Patient Achieving a Durable Complete Response to an Anti-PD-1 Monoclonal Antibody without Developing MG

A 72-year-old man was diagnosed with lung squamous cell carcinoma and had left upper lobectomy and lymph node resection (pathological T2aN2M0 stage IB, PD-L1 tumor proportion score ≥ 50%). He had a 90 pack-year history of cigarette smoking. He received S-1 monotherapy as postoperative adjuvant chemotherapy for two years. However, he was diagnosed with recurrence of lung squamous carcinoma with right cervical and mediastinal lymph node metastases. He had pulmonary metastases and enlargement of the lymph node metastases after receiving four cycles of carboplatin plus nab-paclitaxel as the first-line chemotherapy regimen and one cycle of docetaxel as the second-line chemotherapy regimen. Thus, an anti-PD-1 monoclonal antibody, nivolumab, was selected as the third-line regimen.

Screening tests for autoimmune diseases including disease-specific autoantibodies were done before administration of nivolumab. He had no history of thymic epithelial tumor and autoimmune disease. He had no symptom associated with autoimmune or neuromuscular diseases. His performance status was 0. Creatine kinase was not elevated. However, he was positive for serum anti-AChR Abs (0.8 nM, normal upper limit, 0.2 nM). The potential risks and benefits of an anti-PD-1 antibody therapy for the anti-AChR Ab-seropositive advanced NSCLC patient were carefully evaluated in consultation with neurologists. Then, after obtaining informed consent, nivolumab was administered 3 mg/kg every two weeks with careful monitoring of clinical symptoms and levels of anti-AChR Abs by neurologists. Following four cycles of nivolumab, he had hypothyroidism as an irAE and hormone replacement therapy was initiated. The common irAEs such as pyrexia, rash, interstitial pneumonia, hepatitis, and colitis were not observed. After 17 cycles of nivolumab, a fluorodeoxyglucose (FDG)-positron emission tomography-computed tomography (PET/CT) scan revealed a remarkable shrinkage of metastatic lesions of lung and lymph nodes and he achieved a CR (see Figure 1A,B). Importantly, the patient achieved a durable CR without developing MG even though the levels of anti-AChR Abs were elevated (0.8–1.80 nM) during two years of anti-PD-1 antibody therapy (Figure 1C).

### 2.2. The Tumor of Subclinical MG Patient with a Durable Complete Response to an Anti-PD-1 Antibody Therapy had an Immunologically ‘Hot’ Tumor Microenvironment

To further explore this case, we investigated the immune contexture of pretreatment lung tumor of the anti-AChR Ab-seropositive NSCLC patient who achieved a CR to nivolumab by fluorescent multiplex immunohistochemistry (mIHC). The mIHC analysis has been shown to capture multidimensional data related to tissue architecture, spatial distribution of multiple cell phenotypes, and co-expression of signaling [18,19]. A high density of tumor-infiltrating CD8+ T cells and CD20+ B cells has been shown to correlate with prolonged survival in patients with a wide variety of human cancers including lung cancer [20,21,22]. Regulatory T cells (Tregs) have immunosuppressive activity and play a critical role in maintaining immune homeostasis and negatively regulating anti-tumor immune responses [23,24,25]. Thus, a pretreatment tumor sample from the patient was analyzed for tumor-infiltrating CD8+ T cells, CD20+ B cells, and Tregs (FOXP3+ CD3+ T cells) by fluorescent mIHC. Pan-cytokeratin of tumor cells and PD-L1 were simultaneously stained to evaluate the complex relationship among tissue architecture, spatial distribution of immune cells, and expression of PD-L1.

PD-L1 immunohistochemistry using PD-L1 22C3 pharmDx revealed the tumor PD-L1 tumor proportion score ≥ 50% (Figure 2A,B). CD8+ T cells were infiltrated into both tumor stroma and tumor cell nests (Figure 2). CD20+ B cells were mainly localized to the tumor stroma rather than tumor cell nests and infiltrated at the invasive tumor margin (Figure 3). Tregs were infiltrated into both tumor stroma and tumor cell nests (Figure 4), but the number of tumor-infiltrating Tregs was fewer than conventional T cells (FOXP3-negative CD3+ T cells) or CD8+ T cells (Figure 2 and Figure 4). Altogether, these results demonstrate that an anti-AChR Ab-seropositive NSCLC patient who achieved a CR to nivolumab had an infiltrated–inflamed tumor immune micro-environment and immunologically ‘hot’ tumor [26,27]. The immunologically ‘hot’ tumor micro-environment might associate with the benefits of immune checkpoint blockade therapy without developing MG.

## 3. Discussion

Anti-PD-1 monoclonal Abs block the interaction between PD-1 and its ligand, PD-L1, which unleashes the anti-tumor immune response [1,5,26]. However, the disruption of immune checkpoint signaling can lead to imbalances in immunologic tolerance and result in an unfavorable immune response, which clinically manifest as irAEs [9,11,12,28]. A unique set of inflammatory and autoimmune side effects known as irAEs was quickly recognized in clinical trials in association with the nature of immune checkpoint inhibitors impacting systemic immunity of cancer patients [9,29]. Although the common irAEs are rash, endocrinopathies, interstitial pneumonia, hepatitis, and colitis, rare but serious irAEs have been identified during post-marketing surveillance [8,9,10,11,13]. The pathophysiology underlying these irAEs has not been fully understood, which elucidates mechanisms of irAEs. This is urgently needed to improve their early diagnosis and develop more precise treatments for irAEs [8].

As the use of anti-PD-1/PD-L1 monoclonal Abs is extending to various malignancies with unprecedented speed, there is also an unmet need to identify risks and benefits of immune checkpoint blockade therapy in cancer patients with a history of autoimmune disease [8,29]. Most of the evidence regarding irAEs comes from prospective clinical trials, but cancer patients with concurrent autoimmune disease have been excluded from most of the clinical trials because of concerns that these individuals potentially have an elevated risk for developing serious irAEs. Therefore, the safety of anti-PD-1/PD-L1 monoclonal Abs in cancer patients with a history of autoimmune disease is less clear [8,29]. Recent retrospective studies of immune checkpoint blockade in patients with NSCLC and pre-existing autoimmune disease have shown that adverse events were generally manageable and infrequently led to the discontinuation of immunotherapy. The retrospective studies have also shown that anti-PD-1/PD-L1 monoclonal Abs can achieve clinical benefit in those patients. However, the risks and benefits of immune checkpoint inhibitors in asymptomatic patients with pre-existing disease-specific autoantibodies remain unclear [8,15,16,17,29].

In the current study, we have shown that an anti-AChR Ab-seropositive NSCLC patient achieved a durable CR to an anti-PD-1 monoclonal antibody therapy without developing MG (Figure 1). Makarious et al. showed that the specific MG-related mortality is high (30.4%) in immune checkpoint antibody therapies even though immune checkpoint inhibitor-associated MG is rare [16]. Among the 23 reported cases of irAEs manifesting as MG, 72.7% were de novo, 18.2% were pre-existing MG exacerbations, and only 9.1% (*n* = 2) were exacerbations of subclinical MG (asymptomatic anti-AChR Ab-seropositive cancer patients before administration of immune checkpoint blockade) [16]. One out of the two exacerbations of subclinical MG patients died (the mortality of exacerbations of subclinical MG, 50%). In a study of two-year safety databases based on post-marketing surveys, Suzuki et al. reported that 12 among 9869 cancer patients treated with nivolumab developled MG (0.12%). The nivolumab-induced MG was severe and two MG patients died (MG-related mortality, 17%) [15]. In this study, two cases of exacerbations of subclinical MG have been reported. These studies highlight the importance of recognizing MG as a life-threatening irAE. However, little is known about the potential benefits and the safety of immune checkpoint blockade for subclinical MG [14,15,16].

Understanding the complex tumor microenvironment offers the opportunity to make better prognostic evaluations and select optimum treatments [26,27,30]. Accumulating evidence suggests that a high density of tumor-infiltrating CD8+ T cells and CD20+ B cells strongly associates with positive clinical outcomes in various cancer types [20,21,22,31]. However, the immune contexture of anti-AChR Ab-seropositive tumor response to immune checkpoint inhibitors without developing MG remains unknown. Thus, we analyzed pretreatment tissue of the patient. Infiltrated–inflamed tumor immune micro-environments are considered to be immunologically ‘hot’ tumors and are characterized by high immune infiltrations including CD8+ T cells, B cells, and tumor cells expressing PD-L1 [26,27]. In the current study, the tumor of the subclinical MG patient had high PD-L1 expression and an infiltrated–inflamed tumor immune microenvironment, which suggests similar cases may respond to immune checkpoint blockade therapy without developing MG.

Although anti-PD-1/PD-L1 monoclonal Abs are selectively targeting the PD-1/PD-L1 pathway, the antibodies do not selectively target the PD-1/PD-L1 signaling between tumor antigen-specific T cells and tumor cells. Furthermore, both PD-1 and PD-L1 are expressed not only on effector CD8+ T cells called “killer T cells”, but also on a variety of immune subsets including other T cell subsets and B cells [11,13,32,33,34]. Thus, administered anti-PD-1/PD-L1 monoclonal Abs may bind to the various non-tumor-specific immune subsets and induce the unwanted activation of the immune system, which may disturb the balance established between tolerance and autoimmunity and lead to irAEs such as MG (Figure 5).

A concept of “immune normalization” for the class of drugs called immune checkpoint inhibitors has recently been proposed [1,5]. However, immune checkpoint inhibitors do not always change the immune balance toward a favorable direction for anti-tumor immunity. MG is a B cell–mediated autoimmune disease in which the target auto-antigen is AChR at the neuromuscular junction and also has been known as one of the life-threatening irAEs associated with immune checkpoint blockade for malignancies [14,15,16,35]. PD-1 expresses on activated B cells as well as activated T cells [33,36,37], which indicates that there is a potential risk of triggering B cell–mediated autoimmune disease such as MG by the blockade of the interaction between PD-1 and PD-L1. The evidence suggests that blocking PD-1/PD-L1 signaling may shift the systemic immune balance from the T cell-mediated immune response (cellular immune response) to the B-cell mediated immune response (humoral immune response) [33,36,37] which enhances pre-existing anti-AChR antibody, and may lead to the onset of MG as an irAE (Figure 5A).

CD4+ T cells include T helper type 1 (Th1), which drives the cellular immune responses, and CD4^+^ T helper 2 (Th2), which promotes humoral immune responses. Th2 cells enhance B-cell mediated immunity and promote antibody production [38,39]. PD-1 expresses on Th2 cells as well as Th1 cells and CD8+ T cells. Therefore, the blockade of PD-1/PD-L1 signaling has been shown to promote Th2 cell responses and Th2-type inflammations [13,40], which suggests that immune checkpoint blockade has the potential to modulate the balance between cellular immune response and humoral immune response and may lead to the onset of MG (Figure 5B).

There is no evidence of the safety of anti-PD-1 Ab therapy for cancer patients who are positive for anti-AChR Abs. [15,16]. Although we demonstrated that an anti-AChR-seropositive lung cancer patient had immunologically ‘hot’ tumor and achieved a durable CR to an anti-PD-1 monoclonal antibody therapy without developing MG, our study could not uncover enough evidence to explain the reason why the present case did not develop MG. It is conceivable that the patient might have not been susceptible to an increased anti-AChR antibodies by chance. Thus, clinicians should be cautious to use immune checkpoint blockade for cancer patients with subclinical MG.

Because MG as irAE is life-threatening and closely associated with positive for anti-AChR Ab, the pre-existing serum anti-AChR Ab in cancer patients discourages clinicians from using immune checkpoint inhibitors [14,15,16]. However, the present study indicates that avoiding use of immune checkpoint inhibitors for cancer patients with subclinical MG potentially lead to losing the chance to cure advanced cancers.

## 4. Materials and Methods

### 4.1. Patinet

The Kumamoto University Institutional Review Board approved the study (IRB number, 2287, Approval Date, 23 January 2018).

### 4.2. PD-L1 Staining

PD-L1 expression in the lung cancer specimen was analyzed by immunohistochemical staining using the PD-L1 IHC 22C3 pharmDx antibody (clone 22C3 (Dako North America, Inc., Carpinteria, CA, USA)). The antibody was applied according to DAKO-recommended detection methods. PD-L1 expression in tumor cells was scored as the percentage of stained cells.

### 4.3. Fluorescent Multiplex Immunohistochemistry

Fluorescent multiplex immunohistochemistry was performed with OPAL Multiplex Fluorescent Immunohistochemistry Reagents (PerkinElmer, Waltham, MA, USA) following the manufacturer’s protocol. As outlined in the Table 1, formalin-fixed paraffin-embedded (FFPE) sections were stained by one of the three sequences of primary antibodies, PD-L1, pan-Cytokeratin and CD8, pan-Cytokeratin and CD20, or pan-Cytokeratin, FOXP3, and CD3, respectively, using the tyramide signal amplification (TSA) system with Opal dye reagents. Each labeling step consisted of the following at room temperature. Sections of formalin-fixed, paraffin-embedded tissue were depleted of paraffin and were then hydrated and processed for antigen retrieval by treatment with 10 mM citrate antigen buffers (pH 6.0) via microwave radiation (except for PD-L1 which was processed by pH 9.0 citrate buffer via autoclave). The sections were incubated with 3% H_2_O_2_ for 5 min to inhibit endogenous peroxidase activity, washed with 0.05% Tween in TBS (TBST), exposed to blocking buffer (5% goat serum, 0.5% bovine serum albumin in PBS) for 20 min at room temperature, and incubated for 60 min at room temperature with primary antibodies. They were then washed with TBST, incubated with anti-mouse or anti-rabbit HRP polymer conjugated secondary antibodies (Nichirei, Tokyo, Japan) for 10 min at room temperature except for PD-L1 (incubated for 30 min at room temperature), and washed again, after which immune complexes were detected with Opal reagents. Nuclei were counterstained with 4′,6-diamidino-2-phenylindole dihydrochloride (DAPI) (DOJINDO, Kumamoto, Japan) in water, and whole sections were mounted in ProLong Diamond (Thermo Fisher Scientific, Waltham, MA, USA). Multiplexed slides were observed with a fluorescence microscope (BZ-X700, Keyence, Osaka, Japan). The antibodies used for fluorescent multiplex immunohistochemistry analysis are listed below.

## 5. Conclusions

In conclusion, to the best of our knowledge, this is the first report of an anti-AChR antibody-seropositive cancer patient achieving a durable CR to immune checkpoint blockade therapy without developing MG. This study suggests that immune checkpoint inhibitors can be safely used and provide benefits for advanced cancer patients with an immunologically ‘hot’ tumor even if the anti-AChR antibody are positive. Although careful monitoring clinical manifestation in consultation with a neurologist is needed, immune checkpoint blockade therapy should be considered as a treatment option for asymptomatic anti-AChR Ab-seropositive cancer patients. This study not only provides new insights into the use of immune checkpoint monoclonal Abs for cancer patients with pre-existing disease-specific auto-antibodies, but also may improve our understanding of the pathophysiology underlying irAEs and MG.

## Figures and Tables

**Figure 1 cancers-11-00140-f001:**
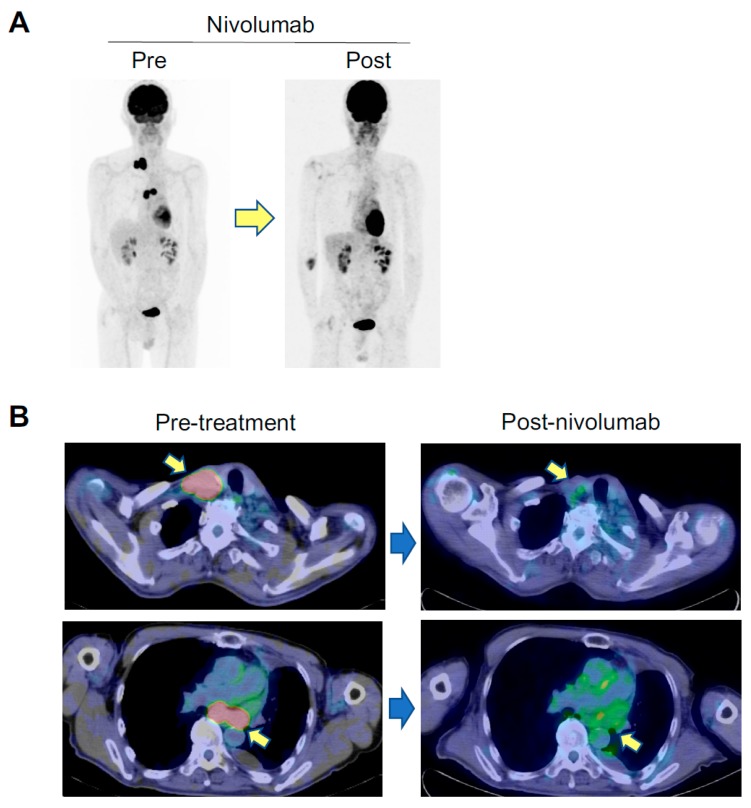
Key imaging and longitudinal analysis of the levels of anti-AChR Abs in asymptomatic anti-AChR Ab-seropositive patient who had a complete response to an anti-PD-1 antibody therapy. Panel (**A**) and (**B**) show FDG-PET/CT imaging pre-nivolumab and post-nivolumab. Arrows in panel (**B**) indicate supraclavicular lymph node (upper panels) and mediastinal lymph node (lower panels) metastases. Panel (**C**) shows the longitudinal analysis of serum concentrations of anti-AChR Ab (nM) before and after nivolumab. The dashed line indicates a normal upper limit of the concentrations of anti-AChR Abs.

**Figure 2 cancers-11-00140-f002:**
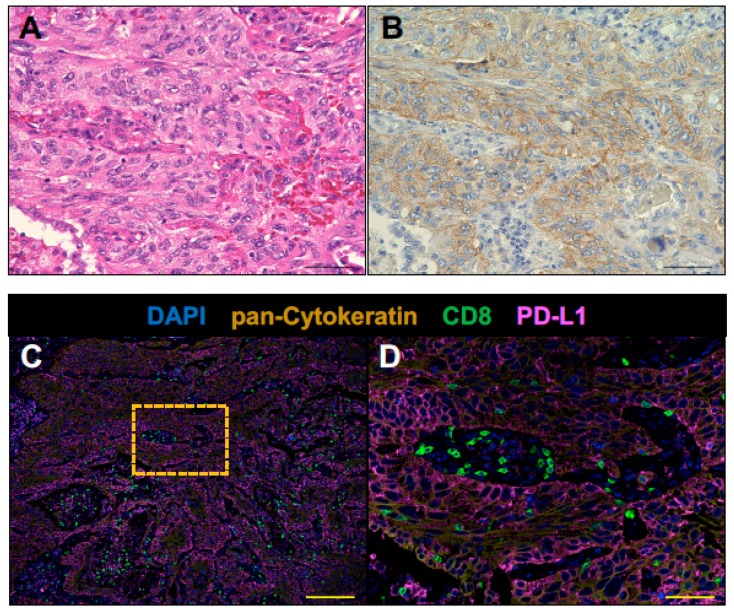
CD8+ T cells infiltrate pretreatment lung tumor of the anti-AChR Ab-seropositive NSCLC patient who achieved a CR to nivolumab. The surgically resected tumor was analyzed by fluorescent multiplex immuno-histochemistry. Serial formalin-fixed paraffin-embedded (FFPE) sections of the tumor sample were stained with Haematoxylin and Eosin (**A**), PD-L1 IHC 22C3 pharmDx (**B**) and analyzed by fluorescent multiplex immunohistochemistry (**C**,**D**). The panel (**D**) shows the boxed region in the panel (**C**) at high magnification. CD8+ T cells (green) were infiltrated into both tumor stroma and pan-Cytokeratin positive tumor cell nests (dark yellow). The tumor expressed PD-L1 (magenta). Nuclei were stained with DAPI (blue). Scale bars, 50 μm (**A**, **B**, and **D**) and 200 μm (**C**), are shown in each panel.

**Figure 3 cancers-11-00140-f003:**
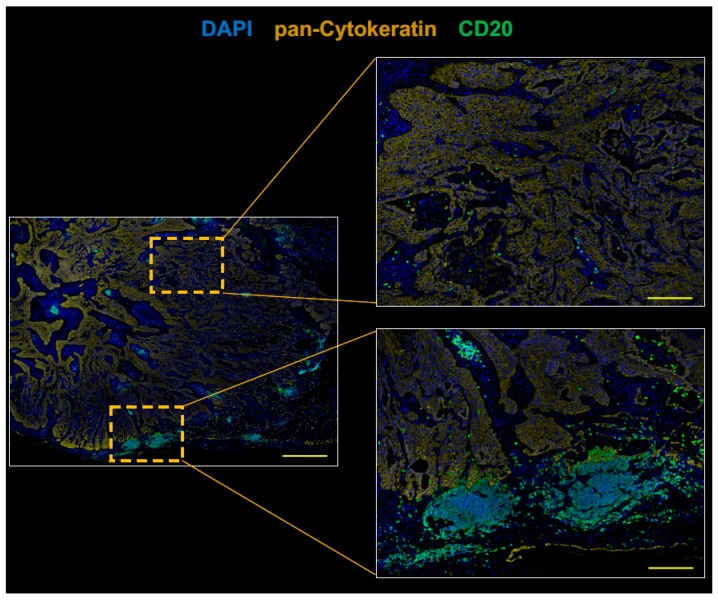
CD20+ B cells infiltrate pretreatment lung tumor at the invasive tumor margin. Serial FFPE sections were stained with antibodies against CD20 (green) and pan-Cytokeratin, and analyzed by fluorescent multiplex immunohistochemistry. The right panels show the boxed regions in the left panel at high magnification. CD20+ B cells (green) were infiltrated at the invasive tumor margin rather than tumor cell nests. Scale bars, 1000 μm (left panel), or 200 μm (right panels) are shown in each panel.

**Figure 4 cancers-11-00140-f004:**
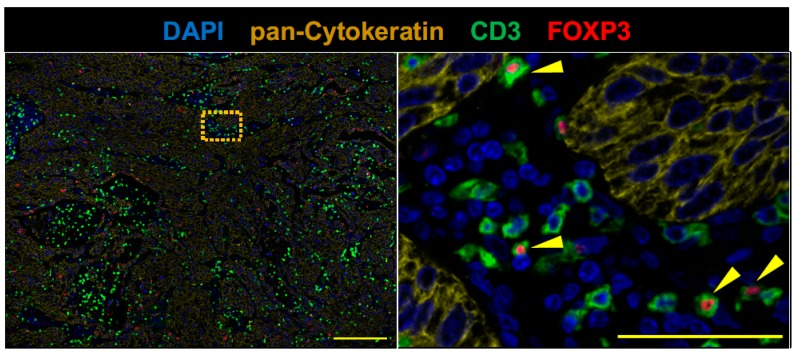
FOXP3+ CD3+ T cells (Tregs) sparsely infiltrate pretreatment lung tumor. Serial FFPE sections were stained with antibodies against CD3 (green), FOXP3 (red), and pan-Cytokeratin. The right panel show the boxed region in the left panel at high magnification. Tregs were sparsely infiltrated in this tumor tissue. Scale bars, 200 μm (left panel), and 50 μm (right panel) are shown in each panel.

**Figure 5 cancers-11-00140-f005:**
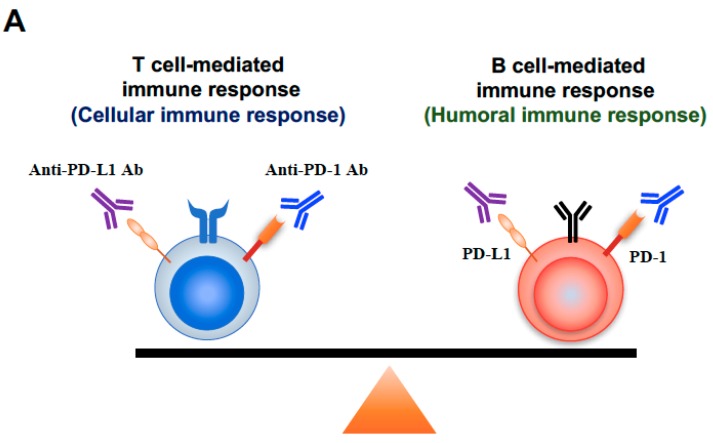
Underlying mechanisms of humoral immune response-associated irAEs. Panel (**A**) shows a model demonstrating the immune balance between a T cell-mediated immune response and a B cell-mediated immune response. Immune checkpoint inhibitors can activate both T cells (cellular immune response) and B cells (humoral immune response), and have the potential to modulate the balance between cellular immune response and humoral immune response, since PD-1/PD-L1 express on both T cells and B cells. Panel (**B**) shows a model demonstrating immune balance between the Th1 cell and the Th2 cell. Immune checkpoint inhibitors can activate both Th1 cells (cellular immune response) and Th2 cells (humoral immune response), and have the potential to modulate the balance between cellular immune response and humoral immune response, since PD-1/PD-L1 express on both Th1 cells and Th2 cells.

**Table 1 cancers-11-00140-t001:** The list of antibodies used for fluorescent multiplex immunohistochemistry analysis.

Figure	Antibody	Clone (Host)/Company	Dilution	Incubation	TSA Dyes
2	CD8	C8/144B (mouse)/Nichirei	undiluted	60 min	520
	pan-Cytokeratin	AE1/AE3 + 5D3 (mouse)/abcam	1:200	60 min	570
	PD-L1	E1L3N (rabbit)/Cell Signaling	1:100	60 min	650
3	CD20	L26 (mouse)/abcam	1:50	60 min	520
	pan-Cytokeratin	AE1/AE3 + 5D3 (mouse)/abcam	1:200	60 min	570
4	CD3	SP7 (rabbit)/abcam	1:100	60 min	520
	FOXP3	236A/E7 (mouse)/abcam	1:100	60 min	570
	pan-Cytokeratin	AE1/AE3 + 5D3 (mouse)/abcam	1:200	60 min	650

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
