# Peer review of "The Risks and Benefits of Immune Checkpoint Blockade in Anti-AChR Antibody-Seropositive Non-Small Cell Lung Cancer Patients"

_cancers, 2019, doi:10.3390/cancers11020140_

Round 1
Reviewer 1 Report
Authors describe the treatment of the patient who has small cell lung carcinoma and was put on nivolumab treatment for the third line regimen. It is an important observation about the specific patient so the paper needs more information about the case study and case history which is missing in the methods section of the paper and abstract.
The authors show the promising data from one patient but still, it is inappropriate to say that this could be a more explored approach for the treatment of more patients. Since the described patient had a substantial level of anti-AChR antibodies to begin with which increased during the therapy, what is the possible explanation for no development of MG? Could it be just a chance factor due to patient being unsusceptible? Also, not all patients develop MG upon current treatment. How can a clinician decide to go for this therapy in such cases. What about other irAEs such as skin related, hepatic, endocrinal etc.? Authors need to include more explanation regarding all these issues.
Author Response
Response to the Reviewers’ comments
We thank the Reviewers for the time they have taken to go over our manuscript and for their helpful suggestions. Specific responses to the Reviewers’ comments follow below.
Responses to the comments from Reviewer #1
Comment 1; Authors describe the treatment of the patient who has small cell lung carcinoma and was put on nivolumab treatment for the third line regimen. It is an important observation about the specific patient so the paper needs more information about the case study and case history which is missing in the methods section of the paper and abstract.
Response to this comment:We thank the Reviewer for this comment. According to the Reviewer’s suggestion, we incorporated the following sentence which includes information of this case; age, sex, chemotherapy-resistance, histology, immune checkpoint inhibitor used for this patient, and the number of prior chemotherapy lines of immune therapy, inthe Method section of Abstractof the revised main text in page 1, lines 22-23.
A 72-year-old manwas diagnosed with chemotherapy-refractory lung squamous cell carcinoma and nivolumab was selected as the third-line regimen.
We also incorporated the following information of this case; history of thymic epithelial tumorand neuromuscular diseases,performance status before immunotherapy, and creatine kinase, inthe Resultsectionof the revised main text in page 2, lines 82, 90-92.
Comment 2; The authors show the promising data from one patient but still, it is inappropriate to say that this could be a more explored approach for the treatment of more patients.
Response to this comment:We agree with the reviewer’s comment. According to Reviewer #1’s suggestion, we omitted the following sentence in Conclusion sectionof the revised main text in page 9.
A deleted sentence: “A larger study would be required to validate our results.”
Comment 3; Since the described patient had a substantial level of anti-AChR antibodies to begin with which increased during the therapy, what is the possible explanation for no development of MG? Could it be just a chance factor due to patient being unsusceptible? Also, not all patients develop MG upon current treatment. How can a clinician decide to go for this therapy in such cases.
Response to this comment:We thank the Reviewer for this comment.According to the Reviewer’s suggestion, we have incorporated the following sentences intothe Discussionsectionof the revised main text on page 9, lines 280-287.
There is no evidence of the safety of anti-PD-1 Ab therapy for cancer patients who are positive for anti-AChR Abs. (ref.15. Neurology2017, 89, 1127-1134, Ref 16. Eur J Cancer 2017, 82, 128-136). Although we demonstrated that an anti-AChR-seropositive lung cancer patient had immunologically ‘hot’ tumor and achieved a durable CR to an anti-PD-1 monoclonal antibody therapy without developing MG, our study could not uncover enough evidence to explain the reason why the present case did not develop MG. It is conceivable that the patient might have been unsusceptible to an increased anti-AChR antibodies by chance. Thus, clinicians should be cautious to use immune checkpoint blockade for cancer patients with subclinical MG.
Comment 4; What about other irAEs such as skin related, hepatic, endocrinal etc.? Authors need to include more explanation regarding all these issues.
Response to this comment: According to the Reviewer’s suggestion, we have incorporated the following sentence intothe Resultsectionof the revised main text on page 3, lines 100-102.
The common irAEs such as pyrexia, rash, interstitial pneumonia, hepatitis and colitis were not observed.

Reviewer 2 Report
This article describes anti-PD-1 (nivolumab) therapy of a patient with non-small cell lung cancer and pre-existing serum antibodies against an acetylcholine receptor. Although these antibodies can be associated with the development of autoimmune disease myasthenia gravis (MG) and their level was increased during 2-year treatment, a durable complete response was achieved without MG development. As immune checkpoint blockade can induce severe immune-related adverse events (irAEs) caused by autoimmune reactions, this case report shows an important observation that demonstrates a possible benefit of nivolumab treatment for patients with pre-existing autoantibodies and thus increased risk of autoimmune disease triggering. Fluorescent multiplex immunohistochemical analysis of the treated tumor found high PD-L1 expression and infiltration with CD8+ cells suggesting “hot” tumor microenvironment that may be necessary for efficient cancer immunotherapy without autoimmunity development.
The authors correctly discuss that the balance between tolerance and autoimmunity can be affected by immune checkpoint blockade, but in Fig. 5 they show the balance between tumor regression and irAEs which is not quite correct as enhanced tumor regression can be accompanied by enhanced autoimmune reactions leading to irAEs. Moreover, B cell-mediated immune response can contribute to tumor regression and the statement “The evidence suggests that blocking PD-1/PD-L1 signaling may shift the systemic immune balance from T cell-mediated anti-tumor immune response (cellular immune response) to B-cell mediated immune response (humoral immune response),…“ is not supported be reference(s). I recommend to modify or omit this part of Discussion and Fig. 5 .
Specific comments:
The name of PD-1 is programmed cell death 1, not anti-programmed cell death 1 (line 37).
Abbreviations FDF-PET/CT (lines 92 and 101) and FFPE (line 131) are not introduced.
PD-L1 antigen was stained by IHC 22C3 pharmDx (missing in Methods) and immunofluorescent staining but results are not described in the manuscript text.
In Fig. 2, the panel D does not show the boxed region from the panel C.
Scale bars in Fig. 2 could be identified by letters A-D (lines 136-137).
FFPE sections were stained with antibodies against CD20 and pan-cytokeratin, not with CD20 and pan-Cytokeratin (line 140). Similar problem is in the Fig. 4 legend (line 146). In this legend, “Figure 3” should be probably substituted by “Tregs” (line 147).
In the line 174, the word “that” or “who” should be omitted.
The sentence in lines 200-202 fits more into Introduction than into Discussion. Moreover, immune checkpoint antibody therapy does not only include targeting the PD-1/PD-L1 pathway.
“Asymptomatic anti-AChR Ab-seropositive cancer” is not appropriate characterization of cancer (lines 32, 252-253).
Whole sections were mounted in ProLong Diamond, not only nuclei (lines 273-274).
The Table is not numbered.
Author Response
Responses to the comments from Reviewer #2
This article describes anti-PD-1 (nivolumab) therapy of a patient with non-small cell lung cancer and pre-existing serum antibodies against an acetylcholine receptor. Although these antibodies can be associated with the development of autoimmune disease myasthenia gravis (MG) and their level was increased during 2-year treatment, a durable complete response was achieved without MG development. As immune checkpoint blockade can induce severe immune-related adverse events (irAEs) caused by autoimmune reactions, this case report shows an important observation that demonstrates a possible benefit of nivolumab treatment for patients with pre-existing autoantibodies and thus increased risk of autoimmune disease triggering. Fluorescent multiplex immunohistochemical analysis of the treated tumor found high PD-L1 expression and infiltration with CD8+ cells suggesting “hot” tumor microenvironment that may be necessary for efficient cancer immunotherapy without autoimmunity development.
Comment 1; The authors correctly discuss that the balance between tolerance and autoimmunity can be affected by immune checkpoint blockade, but in Fig. 5 they show the balance between tumor regression and irAEs which is not quite correct as enhanced tumor regression can be accompanied by enhanced autoimmune reactions leading to irAEs. Moreover, B cell-mediated immune response can contribute to tumor regression and the statement “The evidence suggests that blocking PD-1/PD-L1 signaling may shift the systemic immune balance from T cell-mediated anti-tumor immune response (cellular immune response) to B-cell mediated immune response (humoral immune response),…“ is not supported be reference(s). I recommend to modify or omit this part of Discussion and Fig. 5 .
Response to this comment: We thank the Reviewer for this comment.We agree with the reviewer’s comments. According to the Reviewer’s suggestion, we have modified the Figure 5A-B and revised discussion section regarding Figure 5 in page 8, lines 244-251. Relevant references have also been incorporated.
Revised sentences follow below;
PD-1 expresses on activated B cells, as well as activated T cells [33,36,37], indicating that there is a potential risk of triggering B cell–mediated autoimmune disease such as MGby blockade of the interaction between PD-1 and PD-L1. The evidence suggests that blocking PD-1/PD-L1 signaling mayshift the systemic immune balance from T cell-mediated immune response (cellular immune response) to B-cell mediated immune response (humoral immune response) (Ref. 33. Int Immunol 2013, 25, 129-137,Ref. 36.Annu Rev Med 2014, 65, 185-202, Ref. 37. Proc Natl Acad Sci U S A 2001, 98, 13866-13871), enhancing pre-existing anti-AChR antibody, and may lead tothe onset of MG as an irAE (Figure 5A).
Specific comments:
Comment 2; The name of PD-1 is programmed cell death 1, not anti-programmed cell death 1 (line 37).
Response to this comment: We thank the reviewer for pointing out this error, and have corrected it in the revised manuscriptin page 1, line 39.
Comment 3; Abbreviations FDG-PET/CT (lines 92 and 101) and FFPE (line 131) are not introduced.
Response to this comment: We thank the reviewer forpointing out these errors, and have corrected them in the revised manuscriptin pages 3 and 5.
FDG-PET/CT: fluorodeoxyglucose (FDG)-positron emission tomography-computed tomography (PET/CT)
FFPE: formalin-fixed paraffin-embedded
Comment 4; PD-L1 antigen was stained by IHC 22C3 pharmDx (missing in Methods) and immunofluorescent staining but results are not described in the manuscript text.
Response to this comment: We thank the reviewer for pointing out these error, and have incorporated the following sentences intothe Methodsection of the revised manuscriptin page 10, lines 321-325.
4.2. PD-L1 staining
PD-L1 expression in the lung cancer specimen was analyzed by immunohistochemical staining using the PD-L1 IHC 22C3 pharmDx antibody (clone 22C3 [Dako North America, Inc., Carpinteria, CA]). The antibody was applied according to DAKO-recommended detection methods. PD-L1 expression in tumor cells was scored as
the percentage of stained cells.
We have also incorporated the following sentence intothe Resultsection of the revised manuscriptin page 4, lines 131-132.
PD-L1 immunohistochemistry using PD-L1 22C3 pharmDx revealed the tumor PD-L1 tumor proportion score ≥ 50% (Figure 2A-B).
Comment 5; In Fig. 2, the panel D does not show the boxed region from the panel C.
Response to this comment: We thank the reviewer for pointing out this error, and have revised the Figure 2.
Comment 6; Scale bars in Fig. 2 could be identified by letters A-D (lines 136-137).
Response to this comment: According to the Reviewer’s suggestion, we have revised the relevant figure legend of Figure 2.
A relevant revised figure legend: Scale bars, 50 μm (A, B and D) and 200 μm (C), are shown in each panel.
Comment 7; FFPE sections were stained with antibodies against CD20 and pan-cytokeratin, not with CD20 and pan-Cytokeratin (line 140). Similar problem is in the Fig. 4 legend (line 146). In this legend, “Figure 3” should be probably substituted by “Tregs” (line 147).
Response to this comment: We thank the reviewer for pointing out these errors. We have revised the figure legends of Figure 3 and Figure 4.
Comment 8; In the line 174, the word “that” or “who” should be omitted.
Response to this comment: We thank the reviewer for pointing out this error. We have corrected it in the revised manuscriptin page 7, line 200.
Comment 9; The sentence in lines 200-202 fits more into Introduction than into Discussion.
Response to this comment: According to the Reviewer’s suggestion, we have omitted the relevant sentence because similar descriptions are already in the Introduction section of original manuscript.
Comment 10; Moreover, immune checkpoint antibody therapy does not only include targeting the PD-1/PD-L1 pathway.
Response to this comment: We thank the reviewer for pointing out this error, and have corrected it in the revised manuscriptin page8, line 231.
Comment 11; “Asymptomatic anti-AChR Ab-seropositive cancer” is not appropriate characterization of cancer (lines 32, 252-253).
Response to this comment: We thank the reviewer for pointing out these errors, and have corrected them in the revised manuscriptin page 1, line 34, and in page 10, line 358.
Comment 12; Whole sections were mounted in ProLong Diamond, not only nuclei (lines 273-274).
Response to this comment: We thank the reviewer for pointing out this error, and have corrected it in the revised manuscriptin page 10, line 343.
Comment 13; The Table is not numbered.
Response to this comment: We thank the reviewer for pointing out this error. We have incorporated the following sentence into the revised main text in page 10, line 349.
Table 1.The list of antibodies used for fluorescent multiplex immunohistochemistry analysis.
-
